# Cross Sectional Anatomy and Magnetic Resonance Imaging of the Juvenile Atlantic Puffin Head (Aves, Alcidae, *Fratercula arctica*)

**DOI:** 10.3390/ani13223434

**Published:** 2023-11-07

**Authors:** Marcos Fumero-Hernández, Mario Encinoso, Ayose Melian, Himar Artiles Nuez, Doaa Salman, José Raduan Jaber

**Affiliations:** 1Veterinary Hospital, Faculty of Veterinary Medicine, University of Las Palmas de Gran Canaria, Trasmontaña, Arucas, 35413 Las Palmas, Spain; marcos.fumero101@alu.ulpgc.es; 2Myofauna Servicios Veterinarios, Camino Lomo Grande, Arucas, 35411 Las Palmas, Spain; consulta@myofauna.com; 3IVC Evidensia Los Tarahales, 35013 Islas Canarias, Spain; administracion@hvtarahales.es; 4Department of Animal Medicine, Faculty of Veterinary Medicine, Sohag University, Sohag 82524, Egypt; doaasalman2020@gmail.com; 5Department of Morphology, Faculty of Veterinary Medicine, University of Las Palmas de Gran Canaria, Trasmontaña, Arucas, 35413 Las Palmas, Spain

**Keywords:** anatomical cross-section, magnetic resonance imaging, head, seabirds, Atlantic puffin

## Abstract

**Simple Summary:**

In this study, we described the anatomy of the central nervous system (CNS) and associated structures of the puffin using anatomical cross-sections and advanced imaging techniques such as magnetic resonance imaging (MRI). To the best of the authors’ knowledge, this is the first description of the head in the Atlantic puffin. The results obtained in this study could contribute to future anatomical and pathological studies on related species.

**Abstract:**

The Atlantic puffin is a medium-sized seabird with black and white plumage and orange feet. It is distributed mainly along the northern Atlantic Ocean, and due, among other reasons, to human activities, it is in a threatened situation and classified as a vulnerable species according to the International Union of Conservation of Nature (IUCN). In this study, we used a total of 20 carcasses of juvenile Atlantic puffins to perform MRI, as well as anatomical cross-sections. Thus, an adequate description of the head was made, providing valuable information that could be helpful as a diagnostic tool for veterinary clinicians, who increasingly treat these birds in zoos, rehabilitation centers, and even in the wild.

## 1. Introduction

*Fratercula arctica*, commonly called Atlantic puffin or simply puffin, is a species of seabird belonging to the order *Charadriiformes* and family *Alcidae*. Its scientific name is related to the aspect of its black and white plumage (from the Latin *fratercula*; friar) and its distribution fundamentally throughout the North Atlantic Ocean [1,2]. These seabirds are medium-sized birds that spend most of their lives in pelagic and offshore habitats, as they only come ashore during the reproductive period to nest on cliffs and rocky areas [1,3,4,5]. Males have larger body dimensions than females and are heavier, but their colors are similar (more vivid in the male during mating season). This seabird has a monogamous mating system, reaching sexual maturity around five years of age (range 3 to 6 years) and incubating one egg per clutch [3,6,7,8,9,10,11,12,13]. It feeds mainly on fish, including variable amounts of polychaetes and crustaceans [14,15]. It is the only species of the three that constitute the genus *Fratercula* (along with the horned puffin (*Fraterculacorniculata*) and the tufted puffin (*Fratercula cirrhata*) that is included in the IUCN list of threatened species, being classified as vulnerable [1]. Among the threats these animals face are entrapment in fishing nets [16,17], pollution [7], predation by introduced species [18,19,20], or reduced food availability [21]. Another factor that has shown a tremendous negative impact on puffin populations is climate change, characterized by strong and increasingly common sea storms that can lead to massive strandings [22,23,24,25].

Puffins are essential indicators of the state of marine ecosystems and their changes over time due to their position in the trophic chains because they can act as bioaccumulator organisms [26,27,28]. Therefore, they play a helpful role in the transfer of phosphorus and nitrogen from oceanic waters to the continents [29]. In addition to its biological importance, this charismatic species from the North Atlantic Ocean has become an ecotouristic attraction, and many companies in North America and Europe have developed this type of activity near puffin colonies [3]. For all these reasons, private and governmental associations allocate part of their funds to puffin conservation projects [30,31].

It is considered that birds have a level of brain expansion close to that of mammals. Despite this, research into the neuroanatomy of this taxonomic group has been scarce. However, the irruption of modern imaging techniques such as magnetic resonance imaging (MRI) or computed tomography (CT) has enabled the development of this doctrine [32]. 

In avian medicine, one of the more used imaging diagnostic tools has been conventional radiography, which provides valuable information on musculoskeletal and respiratory processes and alterations of the coelomic cavity (gas, effusions, or masses). However, it is of little use in the head due to the structure overlapping and low resolution [33,34]. In this case, MRI and cross-sectional imaging modalities could provide adequate anatomical and pathological information, avoiding the main drawbacks of radiography [35,36,37,38,39]. It should be noted that, as in human medicine, MRI is the technique of choice for the nervous system study [34], which has already been used in the study of the avian nervous system [35,36,37,38,39,40,41]. Nonetheless, to the authors’ knowledge, no research has been carried out on the nervous system of the Atlantic puffin. Therefore, this study aimed to describe the head of this bird using anatomical cross-sections and MRI, so the results obtained could serve as a basis not only for future research on the puffin but also on other birds or phylogenetically related animals.

## 2. Materials and Methods

### 2.1. Animals

We conducted this study using a cohort of 20 Atlantic puffins, exhibiting a weight ranging from 0.185 to 0.310 kg and a size range spanning 16 cm to 24 cm (measured from beak to tail). These specimens were sourced from the Consejeria de Área de Medio Ambiente, Clima, Energía y Conocimiento of the Cabildo Insular de Gran Canaria. They had perished as a result of mass strandings associated with episodes of severe open sea storms. Most of the specimens were received already deceased, and those that were initially alive but succumbed due to their weakened condition were promptly preserved via freezing for subsequent MRI procedures. These specimens were made available as part of the investigative efforts aimed at elucidating the potential causative factors behind the substantial stranding events. Additionally, the responsible authority granted explicit permission for their inclusion in this study. It is essential to clarify that no animals were intentionally sacrificed or captured for the sole purpose of scientific research.

### 2.2. MRI Technique

The twenty puffins were imaged at Los Tarahales Veterinary Hospital (Las Palmas, Canary Islands, Spain). Imaging data were acquired using a Canon Vantage Elan 1.5 T imaging system, employing the following sequences: T1-weighted (T1W) sequences in the transverse plane (TR: 634 ms, TE: 10, FOV: 1809 × 829, slice thickness: 2 mm, matrix size: 192 × 160), T2-weighted (T2W) sequences in the transverse plane (TR: 4769 ms, TE: 120, FOV: 1809 × 829, slice thickness: 2 mm, matrix size: 192 × 224), T2W sequences in the dorsal plane (TR: 5271 ms, TE: 120, FOV: 1809 × 829, slice thickness: 2.5 mm, matrix size: 240 × 192), and T2W sequences in the sagittal plane (TR: 4450 ms, TE: 120, FOV: 1809 × 829, slice thickness: 2.9 mm, matrix size: 224 × 224). Additionally, enhanced spin-echo sequences were obtained in the dorsal, transverse, and sagittal planes. The resulting MRI images exhibited a slice thickness ranging from 2.7 to 3.5 mm.

### 2.3. Anatomical Sections

Following the completion of the scanning procedure, seven out of the twenty juvenile puffins were placed in a dorsal decubitus position within expanded polystyrene containers and promptly subjected to deep freezing at −80 °C for a duration of 72 h. Subsequently, serial sections of half a centimeter thick were made using an electric band saw to obtain sequential transverse, sagittal, and dorsal anatomical cross sections. These slices were immediately cleaned with water, numbered, and photographed on both sides.

### 2.4. Anatomic Evaluation

Those anatomical cross-sections that better matched the MR images were selected to facilitate the identification of relevant structures of the Atlantic puffin head. In addition, it was necessary to consult textbooks, relevant references on bird anatomy, as well as bone preparations from other seabird specimens [32,33,42,43,44,45].

## 3. Results

In this study, we present a total of 10 figures, which correspond to anatomical sections and T2W MR images. T2W MR images were used because they exhibited higher resolution and contrast. Figure 1 is a sagittal image that shows transverse (A) and dorsal (B) lines, displaying the cross-section levels depicted by roman numerals (II-VIII), which approximately matched the anatomical sections (Figure 2, Figure 3, Figure 4, Figure 5, Figure 6, Figure 7, Figure 8 and Figure 9). These figures are displayed in two images: (A) anatomical cross-section and (B) T2W MR image. In addition, Figure 10 shows a sagittal anatomical section and the corresponding T2W image, identifying the main structures that compose the head of the Atlantic puffin, especially those related to the central nervous system.

### 3.1. Anatomical Sections

The anatomical sections presented in this investigation proved to be valuable for the discernment of various structures constituting the central nervous system and its related components, all of which were annotated in accordance with the *Nomina Anatomica Avium*. Therefore, we observed the avian brain with the telencephalic hemispheres containing the lateral ventricles (see Figure 4A and Figure 7A). Both hemispheres were separated by the *Fissura longitudinalis cerebri* (Figure 6A and Figure 7A). In addition, transverse cross sections were quite helpful in distinguishing a slightly caudolateral disposed rostral groove, the *Vallecula telencephali*, which was presented on the dorsal surface of each hemisphere, as well as a little pointed olfactory bulb situated at the rostral end of each hemisphere (see Figure 2A, Figure 3A and Figure 7A). The *Diencephalon* was identified as a rostral continuation to the *Mesencephalon* and represents the rostral limit of the brain stem (Figure 10A). The dorsal and sagittal anatomical sections allowed the visualization of some components of the *Hypothalamus*, such as the optic chiasm and the optic nerve penetrating the sclera (see Figure 8A, Figure 9 and Figure 10A). Further, these sections allowed us to identify other components of the Atlantic puffin brain, such as the dorsal part of the *Mesencephalon*, the large unpaired median *Corpus cerebelli*, the internal medullary body with an internal white substance, and the little paired cerebellar hemispheres (Figure 4A, Figure 5A, Figure 6A, Figure 7A, Figure 8A, Figure 9A and Figure 10A). Moreover, these sections allowed the identification of the ventral part of the *Rhombencephalon*, including the *pons* and the *Medulla oblongata*, which did not present an obvious demarcation (see Figure 4A, Figure 5A, Figure 6A, Figure 7A, Figure 8A, Figure 9A and Figure 10A).

In addition, dorsal, transverse, and sagittal anatomical sections yielded fundamental insights regarding the avian eyeball, which exhibits a relatively substantial size in proportion to the cranial volume and a lateral positioning characteristic of avian anatomy, assuming a globose morphology with slight medial flattening, as depicted in Figure 2A. Furthermore, the delineation of various ocular components, including the cornea, sclera, retina, vitreous chamber, *Pecten oculi*, and lens, was achieved with precision (see Figure 2A, Figure 3A, Figure 7A, Figure 8A, Figure 9A and Figure 10A). Concurrently, our examination unveiled associated structures about the eyeball, encompassing the interorbital septum, extraocular muscles, and the infraorbital sinus that provides structural support to the eyeball, as illustrated in Figure 2A, Figure 3A, Figure 7A, Figure 8A, Figure 9A and Figure 10A. Moreover, these sections provided essential information about the skull shape and different bony structures comprising the skull, including the nasal, the frontal, the parietal, the pterygoid, and the otic and occipital bones (see Figure 2A, Figure 3A, Figure 4A, Figure 5A, Figure 6A, Figure 7A, Figure 8A, Figure 9A and Figure 10A).

Regarding the nasal cavity, it was found to be bilaterally situated on either side of the median nasal septum. The nares were dorsally positioned at the base of the avian beak, as depicted in Figure 10A. Within the nasal cavity, three nasal concha were discerned in a rostrocaudal sequence: the rostral nasal concha, middle nasal concha, and caudal nasal concha, with particular emphasis on the enhanced development of the middle concha relative to the others (see Figure 7A, Figure 8A and Figure 10A). Moreover, the dorsal, sagittal, and transverse sections were essential to distinguish the roof of the oral cavity and the pharynx (see Figure 2A, Figure 3A, Figure 9A and Figure 10A). Thus, we could observe how the roof was covered by a non-glandular keratinized mucosa forming the transverse ridges (*Rugae palatinae*), which were covered by numerous *Papillae* (see Figure 9A).

### 3.2. Magnetic Resonance Imaging (MRI)

No discernible anatomical distinctions were evident in the examined juvenile puffins. T2-weighted magnetic resonance (T2W MR) images demonstrated a precise alignment with the cranial structures observed in the cadaveric cross sections, furnishing comprehensive insights into the central nervous system (CNS) and its associated structures. Consequently, various constituents of the puffin head’s CNS, the *Bulbus oculi*, and their related elements were adequately differentiated. Thus, in the transverse and dorsal planes, the two telencephalic hemispheres were homogeneous, displaying regions of moderate and hypointense signals corresponding to the cerebral hemispheres and lateral ventricles, respectively (see Figure 3B, Figure 4B, Figure 5B, Figure 6B, Figure 7B and Figure 10B). Notably, these two planes facilitated the identification of the olfactory bulb, which was a small rostrally tapering structure with moderate and homogeneous signal intensity (see Figure 2B, Figure 3B and Figure 7B). The *Hyperpallium*, distinguished by its curved dorsal contour and moderate signal intensity, was another distinct structure identified (see Figure 3B). Conversely, the *Diencephalon*, another forebrain component, displayed limited differentiation from the adjacent *Mesencephalon*, manifesting moderate to low-intensity signals (see Figure 10B). Additionally, prominent features of the *Mesencephalon*, such as the optic lobe, were exclusively discerned in the dorsal T2W MR images, showcasing analogous signal characteristics (see Figure 4B, Figure 8B and Figure 9B). Other essential components of the CNS, including the *Pons* and the *Medulla oblongata*, were identifiable in the transverse, dorsal, and sagittal planes, characterized by low-intensity signals (see Figure 4B, Figure 5B, Figure 6B, Figure 8B, Figure 9B and Figure 10B). Adjacent to the brain stem, the *Corpus cerebelli* and the small paired cerebellar hemispheres exhibited poorly defined regions of hypo-and moderate intense signal (see Figure 4B, Figure 5B, Figure 6B, Figure 7B, Figure 8B, Figure 9B and Figure 10B).

Regarding the *Bulbus oculi*, the vitreous chamber consistently exhibited hyperintense signals across all planes employed (see Figure 2B, Figure 3B, Figure 7B, Figure 8B, Figure 9B and Figure 10B). However, the cornea, sclera, lens, scleral skeleton, and *Pecten oculi* displayed a hypointense signal in the T2W MR images (see Figure 2B, Figure 3B, Figure 7B, Figure 8B, Figure 9B and Figure 10B). The optic nerve showed an accurate visualization using MRI, presenting a hypointense signal, and was surrounded by hyperintense cerebrospinal fluid (see Figure 9B and Figure 10B). Furthermore, the oral cavity, the pharynx, and the trachea were visualized with low-intensity signals in the T2W images (see Figure 2B, Figure 3B, Figure 4B and Figure 5B). Moreover, diverse muscles of the head, including the *Musculus Pterygoideus pars Ventralis, Musculus Tracheolateralis, Musculus Rectus Capitus (Musculus Rectus Dorsalis + Musculus Rectus Ventralis + Musculus Rectus Lateralis)* and *Musculus Constrictor colli* were displayed with intermediate intensity signal in the T2W MR images (see Figure 2B, Figure 6B, Figure 8B and Figure 9B). In addition, various skull bones, including the nasal, the frontal, the parietal, the pterygoid, and the otic and occipital bones, could be identified in the T2W MR images (see Figure 2B, Figure 3B, Figure 4B, Figure 5B, Figure 6B, Figure 7B, Figure 8B, Figure 9B and Figure 10B).

## 4. Discussion

Atlantic puffins can reach latitudes close to the Canary Islands, but their presence on its coasts is uncommon [1]. In this investigation, the specimens employed were sourced from a substantial influx that happened along the coastlines of the Canary Islands in early 2023. The probable cause of this stranding event was attributed to the intense open-sea storms, which resulted in the disorientation and debilitation of the puffins, thereby hampering their foraging abilities. It is noteworthy that severe storms have previously inflicted a considerable adverse impact on puffin populations, culminating in widespread stranding incidents [22,23,24,25].

As a consequence, the avian specimens gathered for our research predominantly comprised juvenile individuals. It posed a constraining element encountered during the course of this study, as the diminutive dimensions of the head (measuring less than 5 cm from the beak to the occipital bone) rendered the acquisition of central nervous system images, with a brain length of less than 1.5 cm, particularly challenging. This challenge was further exacerbated when imaging the rostral aspect of the head that directly pertains to the beak. Furthermore, concerning image resolution, which is evidently more limited than in other species, such as dogs or cats, the physical-volumetric factor of the brains of these animals restricted the achievable resolution even when using a 1.5 T high-field magnet. Other studies performed on the avian brain used 3 T and 4.7 T magnets showed better resolution since they were performed in larger birds, including the red kite, the common buzzard, or the African grey parrot [33,39]. Thus, a higher field magnet and bigger specimens provided better resolution.

Some authors have reported that utilization of various coil types can enhance the signal-to-noise ratio and contrast, both in 1.5 T [46] and higher field intensity equipment [47]. This observation should be considered in future similar studies to improve image resolution. In addition, potential tissue changes associated with post-mortem phenomena should be taken into consideration, as they could also negatively impact the obtained image. However, conducting this type of in vivo study typically involves the use of anesthesia, thereby posing a higher risk to the animals undergoing this procedure [48,49]. The use of anatomical cross sections in different planes allowed the depiction of the normal anatomy of the Atlantic puffin brain and its associated structures with excellent detail, complementing the employ of MRI and enabling the acquisition of valuable anatomical information despite the aforementioned limitations.

In our study, we used magnetic resonance imaging to examine the head of the Atlantic puffin. Analogous anatomical investigations employing advanced diagnostic imaging methodologies have been conducted in this species, yielding encouraging outcomes [50]. To the best of the author’s knowledge, this is the first description of the head of this species using a high-field strength magnet. This technique has already provided essential information in the assessment of the anatomical knowledge of the head and associated structures in other wildlife species, such as reptiles [51,52], rodents [53,54], terrestrial mammals [55,56], as well as different avian species, including red kite (*Milvus milvus*), common buzzard (*Buteo buteo*), and common kestrel (*Falco tinnunculus*) [33], domestic pigeon (*Columba livia domestica*) [41] or African grey parrot (*Psittacus erithacus*) [39], among others. In contrast to conventional imaging procedures, MRI can be used to obtain images via various anatomic planes without repositioning specimens [51,57,58,59]. This technique constitutes a high-value tool to evaluate the central nervous system (CNS) and its associated structures because it provides adequate differentiation between the cranial bones and soft tissue structures compared to other modern imaging techniques. Thus, we use sagittal, transverse, and dorsal MRI planes. These dorsal images were especially helpful in identifying different components of the CNS and soft tissue structures. Nonetheless, it is important to highlight that images could not be accurately interpreted without a thorough knowledge of the tomographic or planimetric anatomy of the subject species.

As previously mentioned in other species, a high field strength magnet was adequate to visualize the avian brain [33]. Hence, we could visualize the *Telencephalon* and the telencephalic hemispheres that were lissencephalic, the olfactory bulb, as well as a dorsal eminence corresponding to the *Hyperpallium*, whose curved dorsal contour was better visualized in the T2W MR images. The identification of this specific structure in this species and the dimensions of the sclerotic ring [60] suggested its strong visual specialization that could be related to the bird’s feeding patterns. In contrast to the anatomical cross sections, the dorsal and transverse T2W MR images were quite helpful in distinguishing the lateral ventricles, which were located in the medial and occipital regions of the cerebral hemispheres until projected laterally towards the olfactory bulb. However, studies performed on birds with higher-resolution equipment did not clearly label the extension of the lateral ventricles [33]. Therefore, we assumed this finding could be attributed to post-mortem changes affecting the ventricular system.

In relation to the *Mesencephalon*, the optic lobes were discernible in both transverse and dorsal cross-sectional views, as well as in corresponding T2-weighted magnetic resonance (MR) images. These structures were situated in lateral proximity to the *Telencephalon*, and their differentiation from the telencephalic hemispheres was evident due to the presence of the tentorial process, which resulted in a distinct demarcation between these regions. Notably, these lobes exhibited substantial volume, primarily attributed to their processing of a significant portion of visual information [45,61].

Similar to terrestrial mammals, the cerebellum of the Atlantic puffin lies above the midsection of the *Mesencephalon*, the pons, and the *Medulla oblongata*, with no discernible clear demarcations between these regions. Examination using transverse cross-sections and T2-weighted MR images revealed the presence of two small cerebellar hemispheres and a larger unpaired central structure. These cerebellar structures were separated by transverse fissures, facilitating a clear distinction between them. Furthermore, these fissures exhibited marked differentiation between the gray and white matter within the cerebellum.

The T2W MR images combined with anatomical cross-sections allowed adequate differentiation of soft tissue structures. Therefore, different ocular components, including the lens, the vitreous and the anterior chambers, the optic nerve, and the sclera, were distinguished. Thus, they appeared with high signal intensity in T2-weighted MR images. However, other formations, such as the *Pecten oculi,* showed low signal intensity and resolution because we used cadaveric specimens, and the intravenous contrast use was not feasible. In vivo studies involving such substances [59] have demonstrated their utility in visualizing these structures in greater detail since their connective tissue scaffold encloses a compacted capillary network.

Despite the low resolution of the bony structures, the skull showed a dome shape, with large orbits separated by a thin interorbital septum and modification of the facial bones to form the beak. The skull bones visualized did not show important pneumatization as happens in other aquatic species. This lower pneumatization of the skull of the puffin [42] could be related to their living and eating habits since, as we have mentioned, they are excellent divers, and therefore, the pneumatization of the skull could be a problem when diving in oceanic waters. In addition to these findings, the ethmoidal labyrinth was not described, probably due to the poor development of the sense of smell, also related to the small rostrally tapering of the olfactory bulb.

## 5. Conclusions

This study presents an initial characterization of the cranial anatomy of the Atlantic puffin utilizing transverse, sagittal, and dorsal magnetic resonance imaging in conjunction with anatomical cross-sections. Magnetic resonance imaging has demonstrated its suitability as a tool, providing comprehensive anatomical insights into the various components comprising the puffin’s cranial region. Despite the high economic cost associated with this equipment, its availability in routine clinical settings remains challenging. Nevertheless, MRI offers valuable information that may serve as a reference in research investigations and clinical evaluations of neurological disorders in seabirds. Subsequent investigations involving adult puffins are essential to facilitate a comparative analysis of cephalic anatomical structures and to explore potential age-related variations in these structures.

## Figures and Tables

**Figure 1 animals-13-03434-f001:**
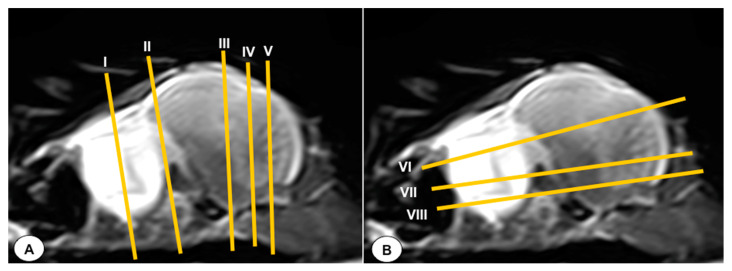
T2-weighted MR sagittal image of the head of an Atlantic puffin (*Fratercula arctica*). The vertical (labeled as (**A**)) and horizontal (labeled as (**B**)) lines correspond to the approximate level of the respective transverse and dorsal slices.

**Figure 2 animals-13-03434-f002:**
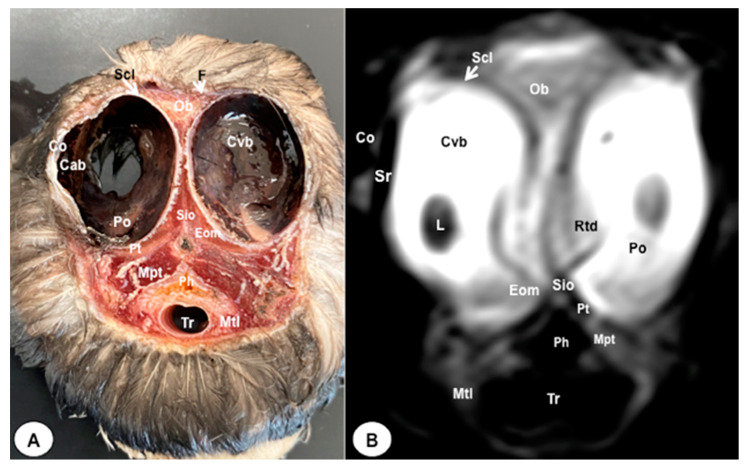
Transverse cross-section (**A**) and T2W MR (**B**) images of the Atlantic puffin head at the level of the *bulbus oculi* corresponding to the line I in Figure 1. Scl: sclera; Sr: sclerotic ring; Cvb: *camera vitrea bulbi*; Co: cornea; Cab: *camera anterior bulbi*: Po: *pecten oculi*; Rtd: retinal detachment; Sio: *septum interorbitalis*; Eom: extraocularmuscle; Ob: olfactory bulb; F: *Os frontale*; Ph: pharinx; Tr: trachea; Pt: *Os pterygoideus*; Mpt: *Musculus pterygoideus pars ventralis*; Mtl: *Musculus tracheolateralis*.

**Figure 3 animals-13-03434-f003:**
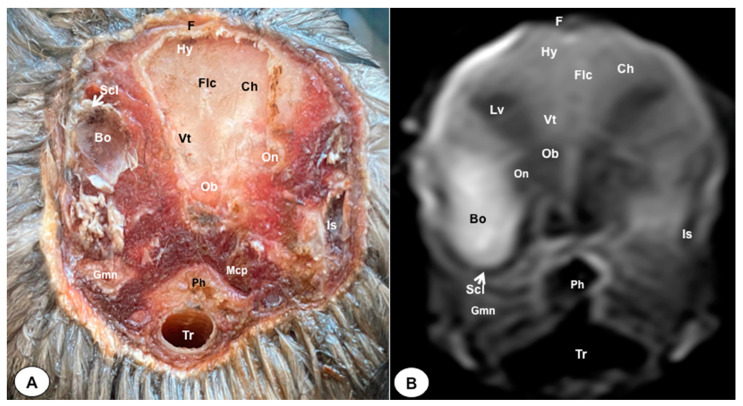
Transverse cross-section (**A**) and T2W MR (**B**) images of the Atlantic puffin head at the level of the olfactory bulb corresponding to line II in Figure 1. F: *Os frontale*; Hy: *hyperpallium*; Ch: cerebral hemisphere; Lv: lateral ventricles; Vt: *Vallecula telencephali*; Flc: *fissura longitudinalis cerebri*; Ob: olfatory bulb; On: optic nerve; Scl: sclera; Bo: *bulbus oculi*;Is: *Sinus infraorbitalis*; Gmn: *Glandula membranae nictitantis*; Ph: pharynx; Mcp: *Musculus constrictor pharyngis*; Tr: trachea.

**Figure 4 animals-13-03434-f004:**
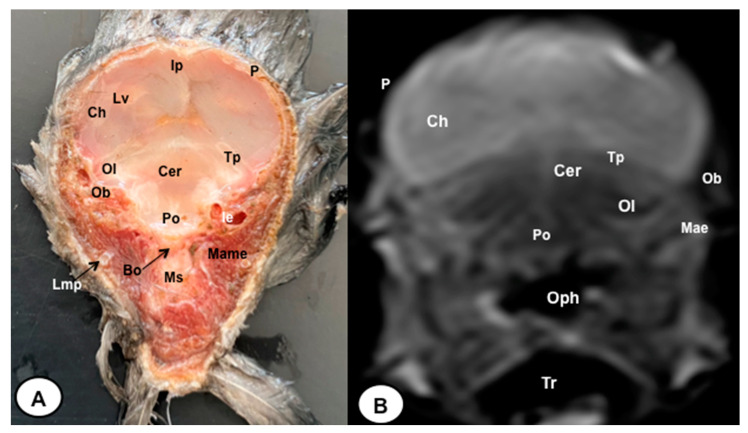
Transverse cross-section (**A**) and T2W MR (**B**) images of the Atlantic puffin head at the level of the optic lobe corresponding to line III in Figure 1. P: *Os parietale*; Ip: interparietal bone; Lv: lateral ventricle;Tp: tentorial process; Bo: *Os basioccipitale*; Ob: otic bones; Ch: cerebral hemisphere; Ol: optic lobe; Cer: *cerebellum*(body); Po: *pons*; Ms: *medulla spinalis*; Mae: *Meatus acusticus externus*; Ie: inner ear; Lmp: Lateral mandibular process; Mame: *Musculus adductor mandibulae externus*; Oph: oropharynx; Tr: trachea.

**Figure 5 animals-13-03434-f005:**
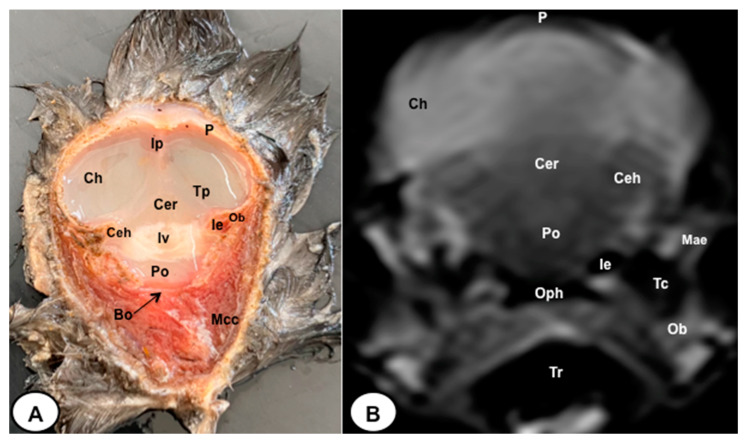
Transverse cross-section (**A**) and T2W MR (**B**) images of the Atlantic puffin head at the level of the pons corresponding to line IV in Figure 1. P: *Os parietale*; Ip: interparietal bone; Tp: tentorial process; Bo: *Os basioccipitale*; Ob: otic bones; Ch: cerebral hemisphere; Ceh: cerebellar hemisphere; Cer: *cerebellum* (body); Iv: fourth ventricle; Po: *pons*; Mae: *Meatus acusticus externus*; Ie: inner ear; Mcc: *Musculus cucularis capitis*; Oph: oropharynx; Tr: trachea.

**Figure 6 animals-13-03434-f006:**
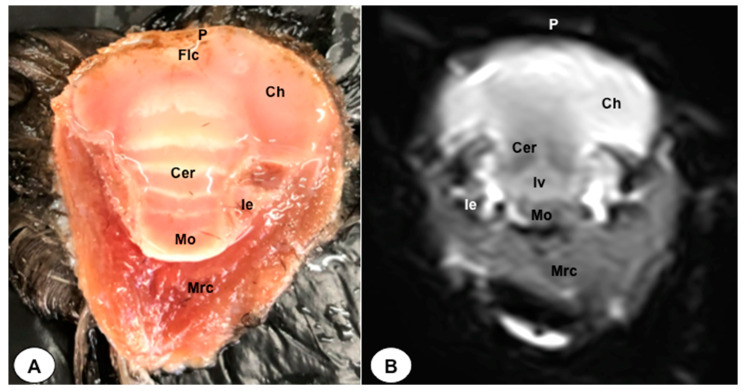
Dorsal cross-section (**A**) and T2W MR (**B**) images of the Atlantic puffin head at the level of the *medulla oblongata* corresponding to line V in Figure 1. P: *Os parietale*; Flc: *fissura longitudinalis cerebri*; Ch: cerebral hemisphere; Cer: *cerebellum*(body); Mo: *Medulla oblongata*; Ie: inner ear; Mrc: *Musculus rectus capitus (Musculus rectus dorsalis + Musculus rectus ventralis + Musculus rectus lateralis)*.

**Figure 7 animals-13-03434-f007:**
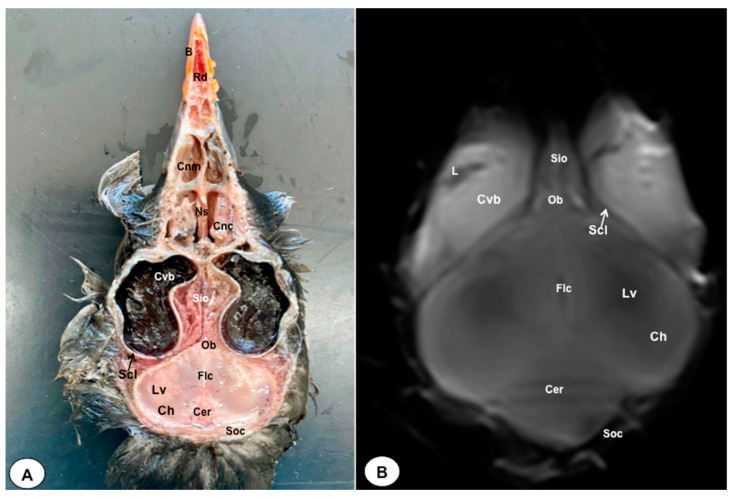
Dorsal cross-section (**A**) and T2W MR (**B**) images of the Atlantic puffin head at the level of the nasal cavity corresponding to line VI in Figure 1. B: beak; Rd: rostral diverticulum; Cnm: *Concha nasalis media*; Cnc: *concha nasalis caudalis*; Ns: nasal septum; Cvb: *camera vitrea bulbi*; Sio: *septum interorbitalis*; Ob: olfatory bulb; Flc: *fissura longitudinalis cerebri*; Ch: cerebral hemisphere; Lv: lateral ventricle; Cer: *cerebellum*; Scl: sclera; Soc: *Os supraoccipitale*.

**Figure 8 animals-13-03434-f008:**
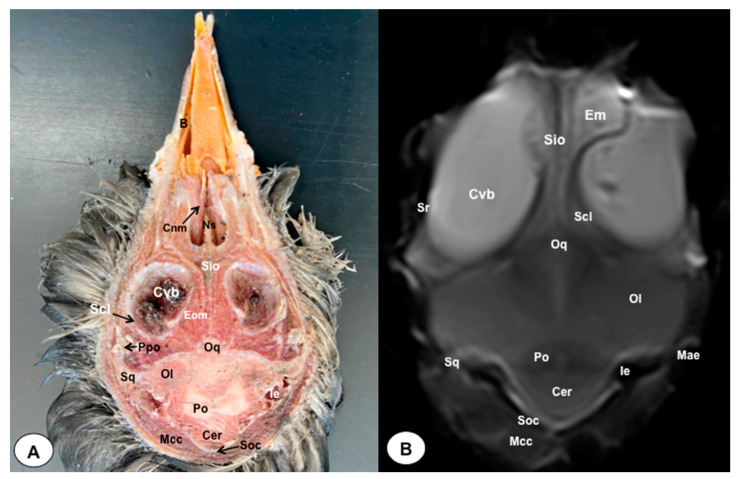
Dorsal cross-section (**A**) and T2W MR (**B**) images of the Atlantic puffin head at the level of optic lobes corresponding to line VII in Figure 1. Sr: sclerotic ring; Scl: sclera; Cvb: *camera vitrea bulbi*; Eom: extraocular muscles; Sio: *septum interorbitalis*; Po: *pons*; Ol: optic lobe; Oq: *chiasma opticum*; Ce: *cerebellum*; Ie: inner ear; Mae: *Meatus acusticus externus*; Sq: *Os squamosum*; Ppo: *Processus postorbitalis*; Soc: *Os supraoccipitale*; Mcc: *Musculus constrictor colli*; Em: *Musculus ethmomandibularis*; Ns: Nasal septum; Cnm: *Concha nasalis media*; B: Beak.

**Figure 9 animals-13-03434-f009:**
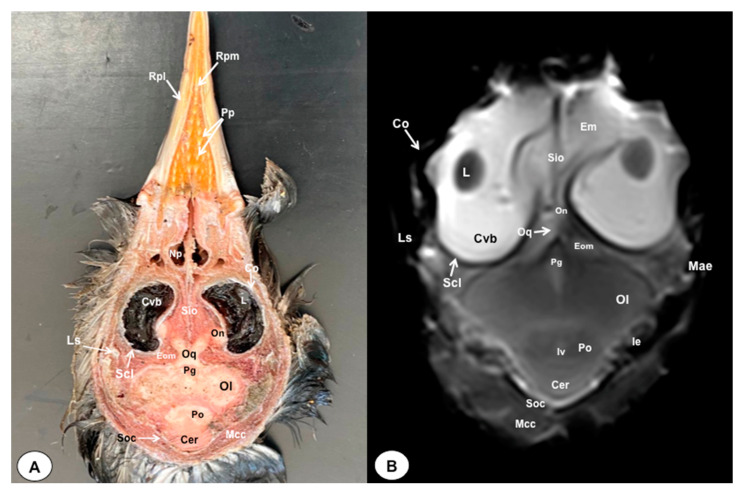
Dorsal cross-section (**A**) and T2W MR (**B**) images of the Atlantic puffin head at the level of the optic chiasm corresponding to line VIII in Figure 1. Cvb: *camera vitrea bulbi*; Sio: *septum interorbitalis*; On: optic nerve; Oq: *chiasma opticum*; Co: cornea; L: lens; Scl: sclera; Eom: extraocular muscle; Ls: *Os laterosphenoidale*; Soc: *Os supraoccipitale*; Po: *pons*; Cer: *cerebellum*; Ol: optic lobe; Pg: pituitary gland; Iv: fourth ventricle; Ie: Inner ear; Mae: *Meatus acusticus externus*; Mcc: *Musculus constrictor colli*; Np: nasopharynx; Em: *Musculus ethmomandibularis*; Cho: *Choanas*; Pp: *Papillae palatinae*; Rpl: *Ruga palatina lateralis*; Rpm: *Ruga palatina mediana*.

**Figure 10 animals-13-03434-f010:**
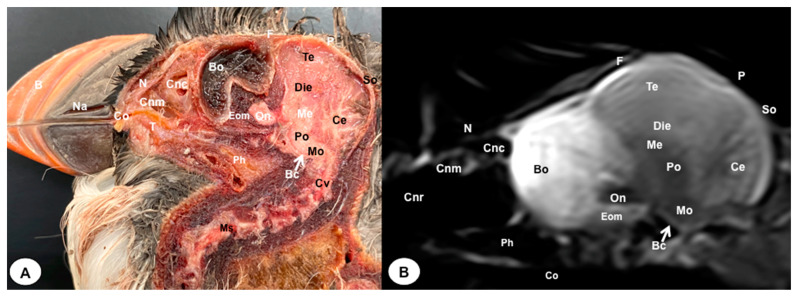
Sagittal cross-section (**A**) and T2W MR (**B**) images of the Atlantic puffin head at the level of the optic nerve. B: beak; Na: *naris*; Co: *cavum oris*; N: *Os nasale*; T: tongue; Cnm: *Concha nasalis media*; Cnc: *concha nasalis caudalis*; Bo: *bulbus oculi*; On: optic nerve; Ph: pharinx; F: *Os frontale*; Te: *telencephalon*; Die: *diencephalon*; Ce: *cerebellum*; Me: *mesencephalon*; Eom: extraocular muscles; Po: pons; Mo: *medulla oblongata*; Ms: *medulla spinalis*; Bc: *basis cranii*; P: *Os parietale*; So: *Os supraoccipitale*; Cv: Cervical vertebra.

## Data Availability

The information is available at “https://accedacris.ulpgc.es”, accessed on 25 June 2023.

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
