# Peer review of "Cross Sectional Anatomy and Magnetic Resonance Imaging of the Juvenile Atlantic Puffin Head (Aves, Alcidae, *Fratercula arctica*)"

_animals, 2023, doi:10.3390/ani13223434_

Round 1
Reviewer 1 Report
Comments and Suggestions for Authors
The Authors describe anatomical structures of the Atlantic puffin by means of MRI in juvenile individuals. The manuscript is clear, well written and scientifically sounds. It could be of interest for ornithologists and veterinary clinicians. I detected only five points (2 major and 3 minor) to be resolved.
1) the work has been made on 20 dead juvenile puffins. The Authors should be discuss the dead reason that is not well clear.
2) Why were they all young? Why did not the authors consider at least some adults for comparison?
3) (minor issue) in the legends of all figures the names of anatomic structures are sometimes uppercase and other times lowercase. Please uniform them.
4) (minor issue) in the reference 40 Psittacus erithacus in italic
5) in the references journal names are sometimes abbreviated with a full stop, other times withouth full stop.
Overall an interesting work that deserves to be published after the suggested improvements.
Author Response
Dear reviewer,
We appreciate your comments regarding our article. The formal defects have been corrected as you suggested. Therefore, we try to respond to points 1 and 2 of your assessment.
1) the work has been made on 20 dead juvenile puffins. The Authors should be discuss the dead reason that is not well clear.
As we mentioned at the beginning of the Discussion section, the specimens used to carry out the study came from a massive arrival occurred in the Canary Islands at the beginning of the year. Among the most probable causes, we included the storms that were generated on that date on the open sea, causing disorientation (remember that it is not common to observe this bird at these latitudes) and weakening of the puffins, and therefore affecting the food search.
2) Why were they all young? Why did not the authors consider at least some adults for comparison?
Since these were not specimens captured for scientific purposes, and as we explained, they arrived by chance on the coast, a specific age range could not be selected. Juvenile specimens are the most affected by this type of storms and the most likely to suffer their consequences. Therefore, young animals were the most abundant among the stranded ones.
3) (minor issue) in the legends of all figures the names of anatomic structures are sometimes uppercase and other times lowercase. Please uniform them.
We have corrected this appreciation as you suggested.
4) (minor issue) in the reference 40 Psittacus erithacus in italic
As you suggested, we have replaced the references in ithalic.
5) in the references journal names are sometimes abbreviated with a full stop, other times withouth full stop.
In the references section, we have adhered to the reference style recommended by the journal. We have placed a period only after the title of journals subject to abbreviation, which explains the presence or absence of period in some of the references.
Reviewer 2 Report
Comments and Suggestions for Authors
In the article Anatomical Cross Sections and Magnetic Resonance Imaging of the Juvenile Atlantic Puffin Head (Aves, Alcidae, Fratercula arctica)." author compare the anatomical slices of the head of an interesting avian species with MRI scan slices.
The information on animals used in the study is too poorly described for a study on the central nervous system. Please provide more information on how long were birds dead before the study, how were they stored etc.
All latin names should be written in italic
The images are low quality and are not diagnostic. Please provide better quality images, since the article is interesting, but those figures are insufficient for publication. Such small structures should be imaged on different type of MR. For reference see this article: https://www.ncbi.nlm.nih.gov/pmc/articles/PMC3784442/pdf/pone.0076135.pdf?fbclid=IwAR3Ffuzu_LYJ8z5m7-n_PElSX8G7uHlkiM4bvRw-LqR5vrvDsKEE8wkaAUM
If it is not possible I would rather see this work as only anatomical slices and description without MRI scans.
Author Response
Dear reviewer,
We really appreciate the comments regarding our article. Corrections have been done, and all latin names have been written in italic, following these recommendations.
The information on animals used in the study is too poorly described for a study on the central nervous system. Please provide more information on how long were birds dead before the study, how were they stored etc.
As we mentioned at the beginning of the Discussion section, the specimens used to carry out the study came from a massive arrival occurred in Canary Islands at the beginning of 2023. Among the most probable causes that the authorities point out are the storms, causing disorientation (remember that it is not common to observe this bird in these latitudes) and weakening of the puffins, and as a consequence, affecting the search for food. We have ruled out, through autopsies, other causes such as Influenza A. The puffins used in our case came to us brought by local fishermen. Most specimens were delivered dead, and those that were alive and ended up dying due to their state of weakness were frozen immediately to perform MRI procedures. No animals were sacrificed or captured for scientific purposes.
All latin names should be written in italic
As you recommend, latin names have been written in italic
The images are low quality and are not diagnostic. Please provide better quality images, since the article is interesting, but those figures are insufficient for publication. Such small structures should be imaged on different type of MR.
We concur with these observations. Nevertheless, the primary objective of this study was to delineate the cranial structures of the Atlantic puffin. Consequently, the dorsal, transverse, and sagittal MR images were helpful in distinguishing relevant components comprising the avian central nervous system. These included the hyperpallium, vallecula telencephali, olfactory bulb, optic nerve, optic chiasm, and optic lobes.
Furthermore, concerning image resolution, which is evidently more limited than in other species, such as dogs or cats, the physical-volumetric factor of the brains of these animals (1.5 cm in diameter) restricted the achievable resolution even when using a 1.5 T high-field magnet. Other studies performed on the avian brain used 3 T and 4,7 T magnet showed better resolution, although they were done in larger birds, such as Red kite, Common buzzard (Stanczyk et al., 2018. 3.0 T MRI anatomy if the central nervous system, eye and inner ear in birds of prey. Vet Radiol Ultrasound) or African Grey Parrot (Fleming et al., 2003. High field strength (4.7 T) MRI of hydrocephalus in an African Greey parrot. Vet Radiol Ultrasound). Thus, the combination of higher field magnet and bigger specimens provided better resolution.
Nonetheless, we will consider the possibility of descriptive analysis using a higher field MRI equipment while being aware that this is the first study performed on this species.
Round 2
Reviewer 2 Report
Comments and Suggestions for Authors
I don't agree with the authors completely on the topic of MRI technique. I work with small animals anatomy and surgery and i use 1,5T MRI on a daily basis, so I am familiar with the capabilities of such a machine. However I find the topic interesting, and this is the first time this species in described. The description of anatomical structure is sufficient and I would like to see this paper published in a present form. I would recommend consulting your MRI settings with a veterinary radiologist who works with MRI. The scans could really be better quality!
Kind regards.